

# Visible and near-infrared radiation may be transmitted or absorbed differently by beetle elytra according to habitat preference

Eva Cuesta[1,2,*] and Jorge M. Lobo[1,*]

[1] Department of Biogeography and Global Change, Museo Nacional de Ciencias Naturales (C.S.I.C.), Madrid, Spain
[2] Escuela Internacional de Doctorado, Universidad Rey Juan Carlos, Móstoles, Madrid, España
* These authors contributed equally to this work.

Corresponding author
Jorge M. Lobo,
mcnj117@mncn.csic.es

## ABSTRACT

**Background:** The exoskeleton of an insect could be an important factor in the success of its evolutionary process. This reaches its maximum expression in beetles, which constitute the most diversified animal taxon. The involvement in the management of environmental radiation could be one of the most important functions of the exoskeleton due to the passive contributions to the thermoregulation of body temperature. We study whether the elytra of two sympatric and closely related beetle species respond differentially to the radiation of distinct wavelengths in agreement with their ecological preferences.

**Methods:** *Onthophagus coenobita* (Herbst) and *O. medius* (Kugelaan) occupy different habitats and environmental conditions (shaded vs. unshaded from solar radiation). The potential adaptive variations to thermoregulation under these different ecological conditions were studied using the responses of their exoskeletons to radiation of different wavelengths (ultraviolet, visible and near-infrared). For these two species, the amounts of the three wavelengths that were reflected, transmitted or absorbed by the exoskeleton were measured using of a spectrophotometer. In addition, the darkness and thickness of the elytra were examined to determine whether these two features influence the management of radiation by the exoskeleton.

**Results:** Both species differ in the management of visible and near-infrared radiation. In agreement with habitat preferences, the species inhabiting shaded conditions would allow infrared and visible radiation to penetrate the elytra more easily to heat internal body parts, while the elytra of the heliophilous species would have increased absorbance of these same types of radiation. An increase in body size (and therefore in elytron thickness) and the quantity of dark spots may serve as barriers against exogenous heat gain. However, the maintenance of between-species differences independent of the effects of these two morphological features led us to suspect that an unconsidered elytron characteristic may also be affecting these differences.

**Discussion:** The results of the involvement of the exoskeleton thickness and spots in the thermoregulation of insects opens new research lines to obtain a better

understanding of the function of the exoskeleton as a passive thermoregulation mechanism in Coleoptera.

## INTRODUCTION

The radiation emitted by the sun can be considered the ultimate cause of the functioning of biogeochemical cycles in nature, and the flux of energy created by this radiation is a decisive force that conditions the behavioral, ecological, morphological, metabolic and physiological characteristics of living organisms (*Hessen, 2008*; *Angilletta, 2009*). This phenomenon is especially true for the animals, such as insects, that depend on radiation and external temperatures to warm their internal parts, thus enhancing metabolic processes and increasing evolutionary rates (*Brown et al., 2004*).

The development of an external skeleton that protects and supports internal body parts is an essential feature of Coleoptera, which is the most diversified animal group on Earth (*Chapman, 2009*) and originated during the early Permian period (*Zhang et al., 2018*). Among the many functions attributed to the exoskeleton of Coleoptera (*Vincent & Wegst, 2004*; *Gorb, 2013*), some authors suggest that its structure and color may help in controlling temperature (*Mikhailov, 2001*; *Ishay et al., 2003*; *Gross, Schmolz & Hilker, 2004*; *Clusella-Trullas, Van Wyk & Spotila, 2007*; *Davis et al., 2008*; *Drotz, Brodin & Nilsson, 2010*; *Roulin, 2014*; *Schweiger & Beierkuhnlein, 2016*). In a series of recent studies carried out on specimens of different species belonging to the Geotrupinae and Scarabaeinae subfamilies (Coleoptera, Scarabaeoidea), the existence of "passive thermoregulation" without associated energetic costs was proposed as a consequence of the interaction of the exoskeleton with different types of electromagnetic radiation (*Carrascal, Jiménez-Ruiz & Lobo, 2017*; *Amore et al., 2017*; *Alves, Hernández & Lobo, 2018*). First, heating experiments under controlled conditions on dried specimens of 13 species of Palaearctic Geotrupidae (*Carrascal, Jiménez-Ruiz & Lobo, 2017*) and seven Neotropical Scarabaeinae species (*Amore et al., 2017*) were carried out. The evidence obtained in these studies suggested that there are interspecific differences in internal body temperatures when these specimens are exposed dorsally to simulated sunlight (*Amore et al., 2017*), and also that these internal temperatures are lower when these specimens are exposed to infrared radiation (*Carrascal, Jiménez-Ruiz & Lobo, 2017*; *Amore et al., 2017*). In a subsequent step (*Alves, Hernández & Lobo, 2018*), a spectrophotometric analysis was used to examine if the reflectance, transmittance and absorbance of the elytra exoskeleton to different wavelengths (ultraviolet, visible and near-infrared) could help to explain the obtained patterns in the internal body heat of the beetles (what can be the main source of body heat?). Thus, analyzing the elytra of five Neotropical Scarabaeinae species of the genus *Canthon* a similar spectrophotometric pattern was obtained: the light from shorter wavelengths is almost entirely absorbed by the elytra, while radiation from longer wavelengths can mostly pass

through the elytra. Consequently, this temperature increase probably results from the transmittance and/or absorbance of non-infrared wavelengths by the dorsal cuticle. The elytra of these species could absorb most of the highly energetic radiation from the ultraviolet and visible parts of the spectrum and convert it into body heat (*Alves, Hernández & Lobo, 2018*; *Pavlović et al., 2018*). All these results suggest that the beetle exoskeleton may allow for the "passive thermoregulation" of body temperatures. As the optima body temperatures of individuals may tend to match the temperatures experienced in the environmental conditions where they occur (*Bozinovic, Calosi & Spicer, 2011*; *Deatherage et al., 2017*), it can be hypothesized that differences in the structure and color of exoskeletons may help explain the ecological and biogeographical characteristics of these organisms, as well as to understand their responses to climatic changes.

In this study, the reflectance, transmittance and absorbance of elytra to different wavelengths of electromagnetic radiation are examined in two phylogenetically close Scarabaeinae species that are locally sympatric and diurnal but differ in their environmental preferences (shaded vs. open habitats). The main aim of this comparison is to verify whether there might be a correspondence between the general environmental preferences of these two species and their capacity to reflect, absorb or transmit radiation of different wavelengths. Specifically, it is hypothesized that species inhabiting shaded conditions would be associated with a higher capacity for elytra transmittance mainly in the infrared range, while heliophilous species should have elytra that are able to cope with a high level of direct sunlight to minimize the risk of overheating. We additionally aim to discern whether these possible interspecific differences in the management of radiation can be attributed solely to differences in body mass and darkness or if, alternatively, any other exoskeletal characteristic could be involved in the thermoregulation mechanism.

## MATERIALS AND METHODS

### Studied species

Individuals of *Onthophagus (Palaeonthophagus) coenobita* (Herbst, 1783) and *O. (Palaeonthophagus) medius* (Kugelann, 1792) were used in this study. The used specimens of these two species were collected within the El Ventorrillo field station with the required permissions provided by the Consejería de Medio Ambiente y Ordenación de Territorio of the Comunidad de Madrid (approval number 10/069528.9/18). The used specimens *O. medius* is a recently proposed taxa (*Rössner, Schönfeld & Ahrens, 2010*) that is very similar to *O. vacca* (Linnaeus, 1767) and can be accurately differentiated by using mitochondrial DNA sequences and, to a lesser extent, by some subtle and overlapping morphological characters among which elytra darkness stands out (*Roy et al., 2016*). In our case, all the studied specimens were carefully selected according to non-overlapping morphological character states (*Roy et al., 2016*), thus the specimens were unambiguously assigned to *O. medius*. These two dung beetle species (Coleoptera; Scarabaeidae) are widely distributed across the Palaearctic region. *O. coenobita* has a geographical distribution ranging from Spain to Sweden and from Belgium to Turkmenistan. Although the knowledge of the geographic distribution of *O. medius* is limited, the available data suggest that *O. vacca* and *O. medius* overlap extensively in their

distributions, although *O. medius* is assumed to be absent in North Africa and present from Spain to Finland and Russia and from Great Britain to Kazakhstan (*Rössner, Schönfeld & Ahrens, 2010*; *Roy et al., 2016*).

Different studies (*Villalba et al., 2002*; *Roggero, Barbero & Palestrini, 2017*; *Rössner, Schönfeld & Ahrens, 2010*; *Roy et al., 2016*) agree on the phylogenetic closeness of *O. coenobita* and *O. vacca* and therefore between *O. coenobita* and *O. medius*. Both species also show clear ecological differences; *O. coenobita* is frequently reported to be associated with forests and shaded localities and feeding on human dung, corpses and mushrooms, in addition to herbivore dung (*Goljan, 1953*; *Jessop, 1986*; *Lumaret, 1990*; *Martín-Piera & López-Colón, 2000*). Despite the lack of reliable data on the environmental preferences of *O. medius*, the available information suggests that this species has a similar ecology to *O. vacca*, which is associated with open green pastures and the consumption of cow, horse or sheep dung, but with the seasonal activity mainly focused on the warmest spring months (*Rössner, Schönfeld & Ahrens, 2010*; *Roy et al., 2016*). A yearly non-published survey conducted during 2017 and 2018 at the "El Ventorrillo" biological station (Madrid, Spain, Lat: 40.75°, Long = −4.02°, ≈1,430 m a.s.l) clearly indicates that these species do not overlap environmentally but may partially coexist seasonally. While *O. medius* and *O. coenobita* do not differ in their general midday daily activity, *O. coenobita* shows a marked preference for woodland sites. Adults of *O. coenobita* are also active at higher air temperatures (approximately 21.8 °C) than those of *O. medius* (approximately 16.8 °C), which is basically due to the early seasonal occurrence of *O. medius* (mean seasonal occurrence around 11th May) compared with *O. coenobita* (mean seasonal occurrence around 7th June).

## Body measurements and spectrophotometric analysis

A total of 10 individuals of each of the two taxa preserved in 70% ethanol were randomly selected from a collection of 4,502 dung beetles belonging to 53 species collected at the "El Ventorrillo" biological station during 2012–2013 (collection deposited in the Museo Nacional de Ciencias Naturales of Madrid). After drying, each specimen was weighed using a Tx423L Shimadzu® balance with a precision of 0.001 g. Subsequently, the left elytron of each specimen was carefully removed with tweezers (Fig. 1) and mounted on black vinyl to estimate their reflectance. The total area of each elytron and the proportion with dark spots were calculated using ImageJ 1.52i software (*Schneider, Rasband & Eliceiri, 2012*; see Fig. 1). The area of the spectrophotometer light beam falling perpendicular to the surface of the elytron was 10.89 mm$^2$, thus the beam practically covered the entire area of the elytron (see Supplemental Data). However, when the area of the elytron was slightly smaller than the area of the light beam, the obtained reflectance measurements were corrected to subtract the part of the light that fell outside of the elytron by using the following equation: RE = RT − (Rv × Av), where RE is the elytron reflectance, RT is the total obtained reflectance, Rv is the reflectance of the vinyl per mm$^2$ for the different wavelengths and Av is the vinyl area not covered by the elytron (10.89-elytron area). In the case of transmittance, such correction is unnecessary because the

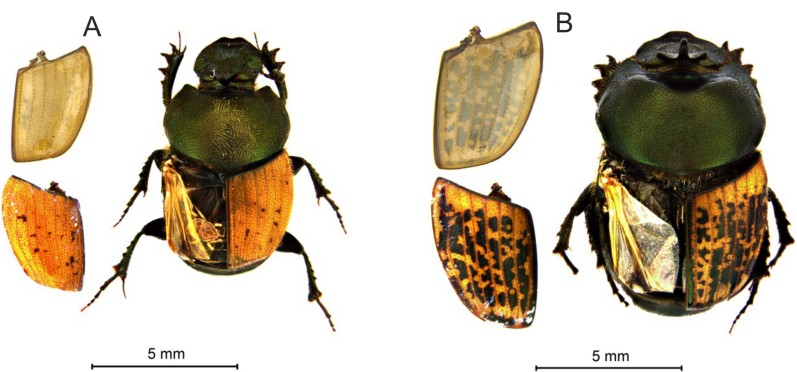

**Figure 1 Habitus and elytra of *Onthophagus coenobita* (A) and *O. medius* (B).** The left elytra was removed, showing colored external and pale internal sides.

elytron is mounted on an opaque metal plate with a hole smaller than the minor size of an elytron (3.301 mm²).

The convexity of the elytra can be considered negligible in both species. The thickness of the left edge of the elytron was also measured with a Nikon Measurescope 10 monocular stereo equipped with a Nikon Digital Counter CM-6S (all measurements in mm). Each elytron was measured on three different occasions by two researchers, and their data were averaged.

Reflectance (R; the return of the electromagnetic radiation from the surface of the elytra) and transmittance (T; the passage of the electromagnetic radiation through the elytra) of the external part of the left elytron (dorsal) were measured with a Shimadzu® UV-2600 spectrophotometer in the wavelength spectrum from 185 to 1,400 nm (at five-nm intervals). This spectrophotometer is equipped with an integrating sphere (ISR-2600Plus) that is able to measure the diffuse and specular reflectance of solid samples. In our case, the measurement conditions of the optical system were adjusted to those needed to measure diffuse reflectance due to the slightly rough characteristics of the elytral surface. Before each spectrophotometer measurement, a white plate of barium sulfate was used to correct the baseline. The obtained data covering the complete wavelength spectrum from 185 to 1,400 nm were divided into three bands; ultraviolet (UV; 185–385 nm), visible (VIS; 390–745 nm) and near-infrared (NIR; 750–1,400 nm). Absorbance (A; the transformation of the electromagnetic radiation received by the elytra into internal energy) was estimated as A = 100 − (T + R) (*Kinoshita, 2008*). Thus, the values of T, R and A were averaged to obtain only one value for each of the three bands as the response variable to determine whether there was variation among elytra in response to the different wavelengths emitted by the sun. The reflectance and transmittance of the internal sides of each elytron (ventral) were also measured but in only the near-infrared range to estimate the possible capacity of the elytra to reflect or transmit body heat generated by beetles. Each measurement was repeated three times by two researchers (2 species × 10 individuals × 2 sides × 3 measurements = 120 measurements for transmittance and reflectance). The three repeated measurements of transmittance and reflectance for each individual were averaged to obtain more stable data that were not dependent on the

position of the elytra or the sector sampled by the spectrophotometer. As the immersion of the elytra in alcohol can modify spectrophotometer measurements (e.g., by eliminating cuticular hydrocarbons), the UV, VIS and NIR reflectance and transmittance values of five fresh elytra of *O. medius* were estimated before and after being subjected to an immersion in 96° alcohol for 16 days. Only the dorsal reflectance in the UV band suggested an effect of alcohol soaking ($t$-test = 2.81, d$f$ = 8; $P$ = 0.02), although the statistical significance of this relationship disappeared when a Bonferroni correction was applied (mean UV reflectance ± SD; fresh elytra = 2.44 ± 0.18; alcohol elytra = 2.70 ± 0.11). If there was a potential effect of the immersion in alcohol on elytra reflectance, we assume here that it was relatively small and similar in the two considered species.

## Statistical analyses

Between-taxa differences in elytron darkness (percentage of the elytron area that was dark) and biometric variables (body mass, elytron area and elytron thickness) were tested by means of Student's $t$-tests considering that these variables follow a normal distribution ($n$ = 10 for each species), and the probability levels were corrected for unequal variances, if applicable. Darkness was considered in these analyses because the melanic compounds responsible for darkness are associated with the absorbance of shortwave radiation and the regulation of body heat (*Pinkert & Zeuss, 2018*). Because the correlations between the three biometric variables were always positive and high (Pearson $r$ values oscillating from 0.83 to 0.96; $P$ < 0.0001 in all cases), elytron thickness was selected for further analyses assuming that a greater elytron thickness could negatively affect the transmittance of radiation toward the interior of the body.

The variation between species (*O. coenobita* vs. *O. medius*) and between elytron sides (internal vs. external; in the case of NIR) in reflectance, transmittance and absorbance (response variables) was examined by ANCOVAs using elytron thickness and elytron darkness as covariates. In these analyses, the explanation of the additive main effects was obviated when two-way interactions showed a relevant effect. Type III sums of squares were used to estimate the partial effect of each explanatory variable once the effects of the other variables were controlled for. The obtained standardized partial regression coefficients can be considered unbiased estimates of the relative importance of predictors, even when they are highly correlated (*Smith et al., 2009*). The effects of the species identity factor, including and excluding covariates, were compared because their change in magnitude and/or sign may indicate the existence of influential confounding or suppressor variables able to overestimate or underestimate the effect of species identity (*Legendre & Legendre, 1998*; *Smith et al., 2009*).

The use of $P$-values as thresholds to discriminate significant and non-significant results is increasingly questioned (*Halsey, 2019*), which is mainly due to their inability to inform about the rate of false positives (*Colquhoun, 2017*). As a consequence, we have abandoned here the use of the terms "statistically significant" and "statistically non-significant," considering $P$-values as indicators of the strength of the evidence of the studied relationships. Thus, Bonferroni corrected $P$-values for multiple comparisons (3 wavelength ranges × 3 response variable; 0.05/9 = 0.006) were considered to identify "strong evidence"

**Table 1 Morphometric values of *Onthophagus coenobita* and *O. medius*.** Mean and standard deviation (SD) of the considered darkness and biometrical variables among *O. coenobita* ($N = 10$) and *O. medius* ($N = 10$) specimens. Student's $t$-tests, corrected for unequal variances, were used to establish statistical differences in these parameters between the two species.

| | O. coenobita | | O. medius | | t | P |
|---|---|---|---|---|---|---|
| | Mean | SD | Mean | SD | | |
| Body mass (mg) | 52.70 | 12.68 | 104.90 | 33.43 | 4.62 | <0.001 |
| Elytral area (mm$^2$) | 8.79 | 1.35 | 13.11 | 1.94 | 5.78 | <0.001 |
| Elytral thickness (µm) | 82.10 | 7.52 | 100.10 | 9.90 | 4.58 | <0.001 |
| % Darkness | 3.68 | 1.83 | 24.59 | 5.20 | 12.00 | <0.001 |

of relationships, while relationships with $P$-values from 0.05 to 0.006 were considered "weak evidence." We checked for homoscedasticity and normality in the residuals of these models. StatSoft's STATISTICA v12.0 was used for these analyses.

## RESULTS

### Biometric and color differences

The average body mass (mg), elytron area (mm$^2$) and elytron thickness (µm) differed between *O. coenobita* and *O. medius*, with higher values for the latter taxon ($P < 0.001$, Table 1). The area of dark pigmentation was also lower for *O. coenobita* than for *O. medius* (Table 1).

### General responses of elytra to wavelength spectrum

The average values of reflectance, transmittance and absorbance across the examined wavelength spectrum for both species and elytron sides (internal and external) are shown in Fig. 2. On average, the reflectance values were lower than the absorbance and transmittance values throughout the complete wavelength spectrum, while absorbance was very high in the ultraviolet and visible wavelength ranges.

### The effect of elytron side

The interaction between species identity and the elytron side factor was highly unlikely to explain the NIR reflectance, transmittance or absorbance (probabilities higher than 0.30 in all cases), indicating that the effect of the elytra side was similar in the two species (Table 2). The dorsal or ventral position of the elytra did not seem to influence the NIR reflectance values, but strong evidence of the influence of elytra position on transmittance and absorbance existed. The ventral NIR transmittance in *O. medius* (56.4%; adjusted means) was higher than that in *O. coenobita* (45.1%) and higher than those measured from the external side of the elytra (52.5% and 37.5%, respectively). Elytra NIR absorbance also seems to be influenced by the elytron side factor (Table 2) as it was lower in *O. medius* than in *O. coenobita* for both the internal side (30.3% vs. 45.8%) and the dorsal side (34.7% vs. 53.5%). Interestingly, the addition of covariates in the regression analyses changed the comparative transmittance and absorbance values of the two species (Table 2). *O. medius* had lower percentages of NIR transmittance and higher percentages of NIR

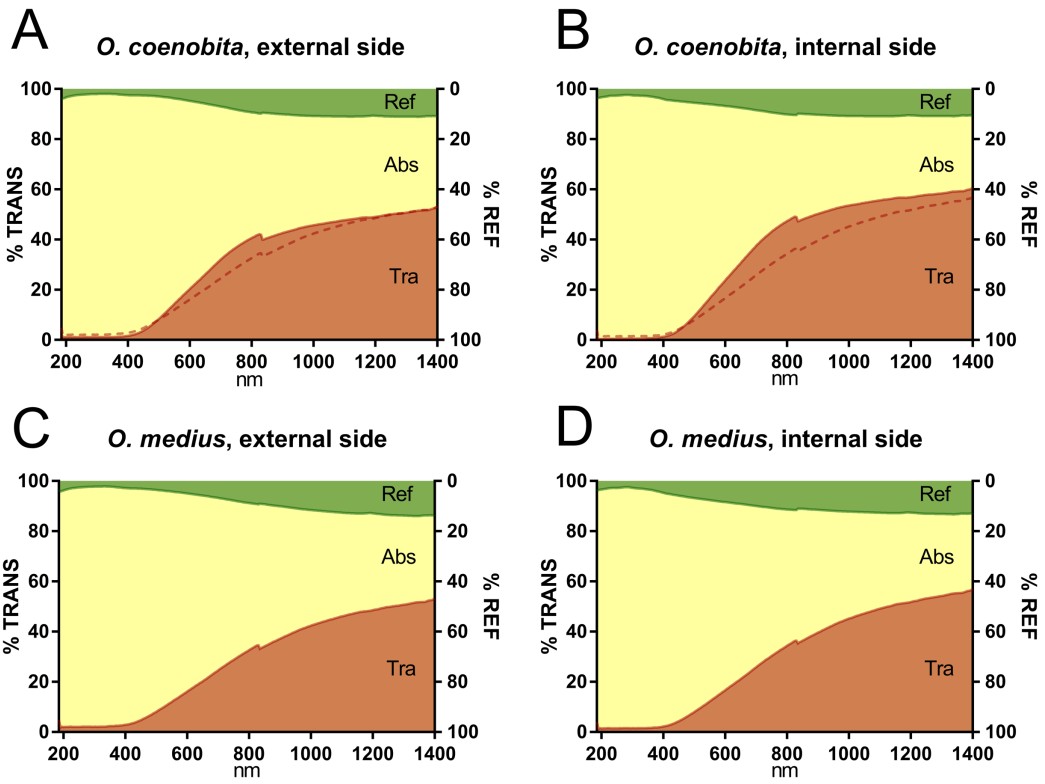

**Figure 2** **Spectrophotometric graphs.** Mean absorbance (ABS), transmittance (TRA) and reflectance (REF) from 185 to 1400 nm of ten individuals of *O. coenobita* (A and B) and *O. medius* (C and D), both for the external (A and C) and the internal sides of the elytra (B and D). The comparison between the two species was facilitated by including a thin broken line representing the transmittance pattern of *O. medius* in the plot of *O. coenobita*. The peak observed at 830 nm is due to the automatic detector change wavelength (the photomultiplier and the InGaAs detector).

absorbance, both dorsally and ventrally, than *O. coenobita* when the raw data were considered. However, this comparative situation was reversed when the effect of elytron thickness and darkness was considered (Table 2).

## The role of thickness and darkness

Our analyses support the existence of strong evidence of the role of elytron thickness in dorsal NIR transmittance and absorbance and weak evidence of the effect of this covariate on NIR reflectance and visible transmittance (Table 2). Thus, the transmittance of the NIR radiation decreased and the NIR absorbance increased when the elytron thickness was higher (according to the signs of the standardized coefficients); a thicker elytron thickness obstructed the penetration of infrared radiation but facilitated its absorbance. The effect of thicker elytra in increasing NIR reflectance and diminishing the transmission of visible radiation should be viewed with caution.

Darkness seems to be the most influential covariate (i.e., highest absolute values of the standardized regression coefficients), showing strong evidence of being an influential variable in explaining the variation in NIR and visible transmittance and absorbance (Table 2). Additionally, the possibility should not be discounted that darker elytra reduce

**Table 2 ANCOVAs results using species identity (*Onthophagus coenobita* and *O. medius*) and elytron side as factors and elytron thickness and elytron darkness as covariates to estimate its effects on reflectance (R), transmittance (T) and absorbance (A).**

| | *O. coenobita* | *O. medius* | Species | Elytron thickness | Elytron darkness | Elytron side | Species × side | $R^2$ |
|---|---|---|---|---|---|---|---|---|
| R-NIR | 10.0/9.0 | 11.8/12.8 | $F_{1,34} = 4.41$ $P = 0.04$ | $F_{1,34} = 5.42$ $P = 0.03$ | $F_{1,34} = 2.95$ $P = 0.10$ | $F_{1,34} = 0.21$ $P = 0.65$ | $F_{1,34} = 0.13$ $P = 0.73$ | 34.12% |
| | | | $\beta = -0.883$ | $\beta = 0.400$ | $\beta = -0.722$ | $\beta = -0.063$ | $\beta = 0.049$ | |
| T-NIR | 46.4/37.5 | 43.5/52.5 | $F_{1,34} = 12.85$ $P = 0.001$ | $F_{1,34} = 29.76$ $P < 0.0001$ | $F_{1,34} = 14.53$ $P = 0.0006$ | $F_{1,34} = 22.74$ $P < 0.0001$ | $F_{1,34} = 2.31$ $P = 0.14$ | 72.61% |
| | | | $\beta = -0.973$ | $\beta = -0.605$ | $\beta = -1.031$ | $\beta = -0.428$ | $\beta = -0.136$ | |
| A-NIR | 43.6/53.5 | 44.7/34.7 | $F_{1,34} = 15.23$ $P = 0.0004$ | $F_{1,34} = 12.36$ $P = 0.001$ | $F_{1,34} = 15.33$ $P = 0.0004$ | $F_{1,34} = 17.35$ $P = 0.0002$ | $F_{1,34} = 1.23$ $P = 0.27$ | 61.14% |
| | | | $\beta = 1.261$ | $\beta = 0.465$ | $\beta = 1.264$ | $\beta = 0.445$ | $\beta = 0.118$ | |
| R-VIS | 4.2/3.1 | 4.7/5.8 | $F_{1,16} = 6.10$ $P = 0.03$ | $F_{1,16} = 0.99$ $P = 0.33$ | $F_{1,16} = 5.63$ $P = 0.03$ | | | 33.78% |
| | | | $\beta = -1.519$ | $\beta = 0.250$ | $\beta = -1.458$ | | | |
| T-VIS | 17.0/9.0 | 14.0/22.0 | $F_{1,16} = 9.69$ $P = 0.007$ | $F_{1,16} = 5.85$ $P = 0.03$ | $F_{1,16} = 12.01$ $P = 0.003$ | | | 60.94% |
| | | | $\beta = -1.470$ | $\beta = -0.467$ | $\beta = -1.636$ | | | |
| A-VIS | 78.8/87.9 | 81.3/72.2 | $F_{1,16} = 11.93$ $P = 0.003$ | $F_{1,16} = 3.92$ $P = 0.06$ | $F_{1,16} = 14.10$ $P = 0.002$ | | | 58.85% |
| | | | $\beta = 1.674$ | $\beta = 0.393$ | $\beta = 1.819$ | | | |
| R-UV | 1.8/1.9 | 2.5/2.5 | $F_{1,16} = 1.63$ $P = 0.22$ | $F_{1,16} = 0.51$ $P = 0.49$ | $F_{1,16} = 0.03$ $P = 0.87$ | | | 59.94% |
| | | | $\beta = -0.611$ | $\beta = 0.139$ | $\beta = 0.078$ | | | |
| T-UV | 1.1/0.4 | 2.2/2.9 | $F_{1,16} = 0.83$ $P = 0.38$ | $F_{1,16} = 0.17$ $P = 0.68$ | $F_{1,16} = 0.22$ $P = 0.66$ | | | 10.02% |
| | | | $\beta = -0.652$ | $\beta = -0.122$ | $\beta = -0.324$ | | | |
| A-UV | 97.0/97.7 | 95.3/94.7 | $F_{1,16} = 1.24$ $P = 0.28$ | $F_{1,16} = 0.09$ $P = 0.76$ | $F_{1,16} = 0.18$ $P = 0.67$ | | | 20.01% |
| | | | $\beta = 0.752$ | $\beta = 0.085$ | $\beta = 0.290$ | | | |

**Note:**
β are the standardized regression coefficients obtained in the regression analyses representing the comparative magnitude and sign of the predictor variables. Results including elytron side and the interaction species × side are only estimated in the case of NIR (β is negative if the average of the internal side is higher than that for the external side). In the case of the species factor β is negative if the average of *O. medius* is higher than that for *O. coenobita*. Those relationships showing *P*-values equal or lower than a Bonferroni corrected *P*-value for multiple comparisons (0.05/9 = 0.006) are considered as "strong evidences" (in underlined bold), while relationships with *P*-values from 0.05 to 0.006 are considered as "weak evidences" (in bold). The two first columns represent average dorsal R, T or A percentages for each species taking into account raw data (first figure) and adjusted means taking into account the effect of the covariates estimated considering that the effect of the covariates is zero in the two species (second figure).

the reflectance of visible radiations. Thus, darker elytra block the passage of infrared and visible radiation, but they favor the absorption of these types of radiation.

## Interspecific differences

Our results provide strong evidence that both species differ in the dorsal transmission and absorbance of NIR and visible radiations and weak evidence of interspecific differences in the dorsal reflectance of these radiation types (Table 2). The elytra of *O. medius* seem to have a higher capacity to prevent the passage of these two types of radiation, while the elytra of *O. coenobita* would better absorb these same types of radiation when the effect of

the studied covariates is considered. Again, the addition of elytron thickness and elytron darkness in the regression analyses reversed the comparative transmittance and absorbance values of the two species (Table 2).

## DISCUSSION

This research aims to assess whether the characteristics of the elytral exoskeleton may contribute to facilitating the thermoregulation of beetles by differentially transmitting, absorbing or reflecting radiation of distinct wavelengths in correspondence with the environmental preferences of the species. The results obtained in this study support this assumption, although more evidence will be needed to clearly discern the extent of the passive role of the beetle exoskeleton in thermal performance. Thus, although the observed disparities can be associated with biometric and darkness differences, the thermal performance of elytra is consistent with the expectations.

Our results agree with those of previous studies (*Carrascal, Jiménez-Ruiz & Lobo, 2017*; *Amore et al., 2017*; *Alves, Hernández & Lobo, 2018*) in that elytron reflectance is minimal, transmittance of infrared radiation is very high, and most of the ultraviolet and visible radiation is absorbed by the elytra. All these exoskeletal characteristics are consistent with the requirements of an ectothermic organism that spends a good deal of time in the soil and would need to obtain body heat from the surrounding infrared and visible radiation (*Pavlović et al., 2018*). Thus, elytra seem to be highly transparent to the heat coming from the sun or the environment but opaque to the most energetic wavelengths capable of causing harmful effects (*Beresford, Selby & Moore, 2013*). Of course, this pattern may vary in those insects exposed to the extreme temperature conditions of deserts in which large parts of visible and near-infrared radiation are reflected (*Shi et al., 2015*). On the other hand, and in agreement with previous results (*Alves, Hernández & Lobo, 2018*), elytral transparency to infrared radiation seems to be slightly higher on the inside part of the elytron than on the outside part, suggesting that the elytra can be slightly more effective at facilitating the removal of body heat in these dung beetle species. In a recent paper, *Pavlović et al. (2018)* demonstrated that short (1,400–3,000 nm) and mid (3,000–8,000 nm) infrared wavelengths, which are mostly absorbed by atmospheric gasses (*Eltbaakh et al., 2011*), can be used to dissipate body heat. Further studies are needed to assess whether the transmittance of these infrared wavelengths is especially high from the internal side of the elytra.

As we expected, interspecific differences in the thermal role of the exoskeleton are clearly mediated by biometric and color characteristics, as exemplified by the effects of elytron thickness and area of dark pigmentation in our analyses. Elytron thickness and especially elytron darkness seem to be particularly relevant in preventing the entry of NIR and visible radiation into the beetle body but also in absorbing these types of radiation. The species with lighter and thinner elytra, which inhabit areas with shaded conditions (*O. coenobita*), would allow these types of radiation to penetrate the elytra more easily to heat their internal body parts. Quite the contrary, the species with darker and thicker elytra (*O. medius*), which inhabited sunny areas, appears to be better able to absorb infrared and visible radiation. These results are in agreement with

those of a recently published study on the thermal capacity of the elytra of the saproxylic beetle *Rosalia alpina* (Linnaeus, 1758) (*Pavlović et al., 2018*), which inhabits the sun-exposed forest along the Euro-Caucasian region. The black patches present in the elytra of this species are also able to absorb visible radiation to heat its body, but the elytra also serve to quickly transmit the infrared radiation to attain thermal equilibrium.

The effects of these physical or physicochemical attributes become so important that they may even reverse the sign of the factor representing species identity. This statistical result has been long recognized and is especially frequent when dealing with correlated predictors (*Leamer, 1975*). In biological and environmental data, the use of non-independent explanatory variables is the norm rather than the exception and not including a valuable predictor because it is correlated with others may imply under or overestimations of the effects of the considered predictors (*Smith et al., 2009*). In our case, the inclusion of covariates in the models reversed the comparative transmittance and absorbance values that could be obtained for the two considered species. Therefore, as the species identity factor continued to be relevant when elytron darkness and thickness were considered, it could not be excluded that some additional and unknown morpho-structural differences may also be relevant in explaining the detected interspecific differences in the capacity of the elytra to manage radiations. Notwithstanding the above, caution is required when determining the comparative roles of correlated features such as elytron thickness and darkness. Additional studies are thus needed to cover a broad range of species with different degrees of darkening and elytron thicknesses to better discriminate the comparative roles of biometric and color characteristics on the thermal performance of the beetle elytral cuticle.

The maintenance of the strength of the exoskeleton with the increase in body size may imply increasing thickness both allometrically and isometrically (*Evans & Sanson, 2005*; *Lease & Wolf, 2010*). Thus, an increase in the body size of dung beetles can provide extra advantages in open habitats by avoiding the internal overheating of the body under sunny conditions. Similarly, the darkening of the exoskeleton could be partially considered an evolutionary strategy to diminish heat transmission into the body. This supposition collides with the thermal melanism hypothesis, which predicts that a darker color may be advantageous in colder environments (*Kalmus, 1941*; *Schweiger & Beierkuhnlein, 2016*; *Galván, Rodríguez-Martínez & Carrascal, 2018*) but could explain why desert beetles are often dark (*Turner & Lombard, 1990*). In our case, the darkest elytra seem to make the access of infrared and visible radiations into the body more difficult, also facilitating the absorbance of these types of radiation; however, the elytra do not influence the management of UV radiation. As the transmittance and absorbance of solar radiation may vary between the elytral parts with different colors and structures (*Pavlović et al., 2018*), the future use of microspectrometry will be recommended to further assess the specific role of black patches in thermal balance.

As in the case of body size, the reduced transmittance of near-infrared radiation by the dark specimens of our two considered species may be a strategy to avoid overheating under some circumstances. Considering that more than 50% of the

total sunlight incident energy corresponds to this wavelength spectrum (*Stuart-Fox, Newton & Clusella-Trullas, 2017*), the management of near-infrared radiation by the beetle exoskeleton should be considered. In this case, darkness can affect thermal performance due to its effect on longwave radiation, which is invisible to the human eye (*Stuart-Fox, Newton & Clusella-Trullas, 2017*). However, *Pantelić et al. (2017)* highlight the low capacity of melanin to absorb the infrared radiation in a dusk moth species, which could indicate that the structural component in which the pigment is embedded could influence this response. Additional experiments are needed to better estimate whether the elytra of different beetle species differ in their capacities to manage distinct wavelengths and discern the comparative roles of body size and darkness in the thermoregulation of beetles.

## CONCLUSIONS

The main hypothesis of this research has been that the elytra of two closely related beetle species will manage environmental radiation differentially in agreement with their contrasting environmental preferences. Thus, the results are in line with what was expected because the elytra of the species inhabiting areas under shaded conditions (*O. coenobita*) allow the entry of infrared and visible radiation more easily, while the heliophilous species (*O. medius*) would better absorb these same types of radiation. These differences are determined largely by the thickness and darkness of the elytra, but we cannot rule out the role that other unknown factors could play in these differences. Further and more comprehensive studies are needed to corroborate the role of the elytral exoskeleton as a mechanism of "passive thermoregulation."

## ACKNOWLEDGEMENTS

We are indebted to Luis María Carrascal for his valuable suggestions. The comments provided by three anonymous referees have improved this manuscript.

### Funding

This work was supported by the MINECOFEDER Project CGL2015-64489-P and the MINECO-FEDER Contract BES-2016-077087 granted to the Eva Cuesta. The funders had no role in study design, data collection and analysis, decision to publish, or preparation of the manuscript.

### Grant Disclosures

The following grant information was disclosed by the authors:
MINECOFEDER Project: CGL2015-64489-P.
MINECO-FEDER Contract: BES-2016-077087.

### Competing Interests

The authors declare that they have no competing interests.

## Author Contributions

- Eva Cuesta conceived and designed the experiments, performed the experiments, analyzed the data, contributed reagents/materials/analysis tools, prepared figures and/or tables, authored or reviewed drafts of the paper, approved the final draft.
- Jorge M. Lobo conceived and designed the experiments, performed the experiments, analyzed the data, contributed reagents/materials/analysis tools, prepared figures and/or tables, authored or reviewed drafts of the paper, approved the final draft.

## Field Study Permissions

The following information was supplied relating to field study approvals (i.e., approving body and any reference numbers):

Consejería de medio Ambiente y Ordenación del Territorio, Comunidad de Madrid is charged to provide collection permissions for not protected animals as those used in this study (Ref: 10/069528.9/18). All these specimens were collected within a field station of the MNCN (El Ventorrillo biological station).

## Data Availability

Raw measurements are available as a Supplemental File.

## Supplemental Information

Supplemental information for this article can be found online at http://dx.doi.org/10.7717/peerj.8104#supplemental-information.

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
