# Peer review of "Visible and near-infrared radiation may be transmitted or absorbed differently by beetle elytra according to habitat preference"

_PeerJ, doi:10.7717/peerj.8104_

## Round 0.1 · original submission · Major Revisions

Dear Drs. Cuesta and Lobo:

Thanks for submitting your manuscript to PeerJ. I have now received three independent reviews of your work, and as you will see, the reviewers raised some concerns about the research. Despite this, these reviewers are optimistic about your work and the potential impact it will have on research communities studying the ecology and evolution of beetles. Thus, I encourage you to revise your manuscript, accordingly, taking into account all of the concerns raised by both reviewers.

Your revision should concentrate on making the text more clear and easier to follow. Specific examples are provided by the reviewers. Please also ensure that important background information is provided as well as all relevant references.

While the concerns of the reviewers are relatively minor, this is a major revision to ensure that the original reviewers have a chance to evaluate your responses to their concerns.

I look forward to seeing your revision, and thanks again for submitting your work to PeerJ.

Good luck with your revision,

-joe

Reviewer 1 ·

Basic reporting

The manuscript is written in a clear, unambiguous, technically correct English, and is conformed to the professional standards of courtesy and expression.
All the literature references are relevant, and correcly listed. Introduction furnished sufficient field background/context to demonstrate how the research fits into the broader field of knowledge.
The structure of the article conforms to the suggested format of PeerJ. The figures are relevant to the content of the article, and appropriately described and labeled, their resolution is sufficient. All appropriate raw data have been made available in accordance with the journal data sharing policy.
The submitted paper is “self-contained”, and can be regarded as an appropriate ‘”unit of publication” being all the included results relevant to the hypotheses.

Experimental design

The research described in the manuscript is surely included within the aims and scope of the journal. The research question was well defined, relevant & meaningful. The research examined some interesting mechanism of insect thermoregulation filling an identified knowledge gap. The technical standard of the present research is high and have been conducted rigorously in accordance with the prevailing ethical standard in the field. The methods were described with sufficient detail and information to be reproducible by another investigator.

Validity of the findings

The present research explore new and interesting aspects of insect thermoregulation, which could have a larger impact on studies of insect physiology. The provided underlying data are robust, statistically sound, & controlled. The data on which the conclusions are based are provided in the Supplementary Material section. The results confirmed the hypothesis, and the conclusions are well stated, linked to original research question and limited to supporting results.

Additional comments

The paper is surely interesting and the experimental design is correct. Hovewer, I have some comments for the authors:
I'm wondering if the potential presence of "ultraviolet" dots on the elytral surface was tested.
The exact portion of the elytron used in the spectrophotometric analysis and thickness measurement must be showed in figure.
Since a low number of individuals was here examined, was the elytral colour variability (i.e., more or less extended dark dots) tested for the two species? The research was limited to two species, thus the results have to be confirmed using a larger dataset.

rows 56-61. the phrase is unclear, please modify it
rows 62-64. please explain better the problem of "limited evolvability" and "huge diversification"
rows 64-67 perhaps the phrase should be modified. The authors defined the Coleoptera as the most diversified animal group (this could likely be) but without explaining the reasons why. It is also unclear how the high diversification should be related to the thermoregulation mechanism. The presence of thickened forewings is a rather common pattern in insects (e.g., Dermaptera, Orthoptera and so on), although usually not so sclerotized as in Coleoptera. The presence of thick forewings cannot be regarded as a peculiar aspect exclusive to Coleoptera.
row 97. the phrase could be modified into "any other exoskeletal character could be involved in the thermoregulation mechanism".
row 117. According to the authors, the O. medius specimens were not identified using DNA barcoding but only by morphological characters. In row 115 the very same characters were defined "subtle and overlapping". Was the rate of incorrect identification tested before the research?

Reviewer 2 ·

Basic reporting

The manuscript analyzes spectrophotometric and biometric characteristics of the elytra of the two related insect species, with the addition of certain statistical analyzes of the obtained parameters. I found the idea interesting, but several aspects and implications could be improved and some points need the attention of the authors.
- The use of the English language needs to be improved. It would be the best to consult a native speaker colleague or some language service. The authors should pay attention to the length and construction of the sentences, especially in the introduction part.

- The authors cite two papers previously published by them:
Alves VM, Hernández MI, Lobo JM. 2018. Elytra absorb ultraviolet radiation but transmit NIR radiation in Neotropical Canthon Species (Coleoptera, Scarabaeinae).
Photochemistry and Photobiology 94:532-539.
Amore V, Hernández MIM, Carrascal LM, Lobo JM. 2017. Exoskeleton may influence the internal body temperatures of Neotropical dung beetles (Col. Scarabaeinae). PeerJ
5:e3349.
These two papers have similar hypotheses and methodology with the submitted manuscript, only other insect species were analysed. I would like the authors to discuss this prior work of theirs and to state how their current manuscript advances beyond or correlate with this prior work.

Experimental design

- In the first two lines of Materials and methods (101,102) authors say that studied specimens are Onthophagus coenobita (Herbst, 1783) and Onthophagus vacca. Later in the text they explain that O. medius was studied and not O. vacca – ask them to synchronize the text.
- I suggest shortening the text about O. vaca, it’s not relevant for the study, or to move it into discussion. Also, lines from 118-135, about ecological characteristics of the species, should be displaced and analysed in discussion section. Here, give some basic morphological description of the species that initially attracted your attention (like, O.coenobita has yellow elytra and O. medius yellow with dark spots).
- How did they measure the weight of the specimens?

Validity of the findings

- Paragraph form lines 242-245 “The transmittance of the NIR radiation significantly decreased when elytron darkness and thickness were higher, obstructing the penetration of infrared radiation. Darkness seems to be the most influential parameter in this case (i.e., highest absolute values of the standardized regression coefficients).”- darkness is due to pigments, mostly melanin, that has absorption spectrum with an exponential decrease from the UV to the IR part of the spectrum (Pantelić et al., 2017, Physical Review E, 95(3), 032405, not cited), thus it should increase absorption in the visible and UV part of the spectrum, without special impact in IR. I would like authors to explain and discuss this.
- The previous comment also relates to the statement from lines 276-277 (“...while elytron darkness seems to be especially relevant in preventing the entry of NIR radiation
into the beetle body”)
- 259-256 “This research aims to assess if the characteristics of the elytral exoskeleton may contribute to facilitating the thermoregulation of beetles by differentially transmitting, absorbing or reflecting radiation of distinct wavelengths in correspondence with the environmental preferences of the species.”- Please be careful with this kind of statements, because you haven’t done any thermal measurements or thermal exchange mechanisms analyses, but only spectrophotometric and biometric analyses. Thus, this is not the primary aim of the manuscript, please rephrase it.
- 295-301 “This supposition would contradict the thermal
melanism hypothesis, which predicts that darker colour may be advantageous under colder
environments (Kalmus, 1941; Schweiger and Beierkuhnlein, 2016, Galván et al. 2018), but could explain why desert beetles are often dark (Turner & Lombard, 1990). In our case, the darkest elytra seem to make the access of infrared radiation into the body more difficult but do not seem to influence the absorbance of visible and UV radiation, unlike other situations that have been studied.”- Too bravely and freely interpretation without clear evidence, see first comment from this section.

- One very important statement is that authors have measured transparency of the elytra in the near IR part of the spectrum (1400 nm), but insect cuticle shows two strong IR transmission minima at approximately 3 and 6 µm (Pavlovic et al., 2018, Journal of thermal biology, 76, 126-138, cited). It would be helpful if authors pointed out to this and discussed it or to extend their measurements to this part of the spectrum.

Reviewer 3 ·

Basic reporting

The paper is well written and present novel and relevant results on the physical mechanisms responsible for the management of environmental radiation by the exoskeleton of these beetles. I have only some specific comments, all of them related to their criteria for considering a P value statistically significant.

Experimental design

no comments

Validity of the findings

Please, see my general comments.

Additional comments

This study analyzes whether the elytra of two sympatric and close related beetle species (Onthophagus coenobita and O. medius) respond differentially to the radiation of distinct wavelengths in agreement with their ecological preferences. The paper is well written and present novel and relevant results on the physical mechanisms responsible for the management of environmental radiation by the exoskeleton of these beetles.

I have only some specific comments.

Line 194. Change “are” to “were”
Lines 221-222. I do not believe that “UV reflectance values are slightly higher in the specimens of O. medius than in O. coenobita (0.53% vs. 0.27%). The value of O. medius, 0.53%, is almost twice the value of O. coenobita, 0.27, so it is clearly higher.
Line 224. If t=2.56, P=0.01, you cannot say: “they tended to vary”, they significantly varied, but, please, explain in which way lower or higher? And also explain what was your criteria for considering a P value statistically significant. P ≤0.05 is generally considered statistically significant. Check your owner lines 194-195 “When these between-species comparisons are statistically significant (P≤0.01)…”
Lines 224-225. In the same vein, in the case of Elytron position, you find significant differences, so you cannot say: Elytron also seemed to…
Line 227. Eliminate “seems to be” and replace by: is
Line 239. P=0.02 is statistically significant when you decide that the significance level is less than or equal to 0.05 (in the case that you consider this alpha level as significant), why do you decide to consider this P value as marginally significant?
Lines 259-262. This sentence is too long. A period at the end of line 259 “expectation. “solves this problem.

---

## Round 0.2 · Minor Revisions

Dear Drs. Cuesta and Lobo:

Thanks for revising your manuscript. The reviewers are very satisfied with your revision (as am I). Great! However, per reviewer 2, there are a few minor issues to consider. Please address these ASAP so we may move towards acceptance of your work.

Best,

-joe

Reviewer 1 ·

Basic reporting

The English is clear, and correctly used in the manuscript
The references are enough, the background is properly discussed.
The structure of the manuscript is correct, figures are of good quality, tables are clear.
The results are interesting, and support the hypothesis

Experimental design

The design of the present research is surely original.
The research questions are clearly defined, the identified research gap was filled by the present study
the investigation was correctly done, in conformity with the ethical standards in the research field.
The methods were destribed in details, and can surely be replicated

Validity of the findings

The research is inttresting, and new results are given.
The dataset is robust and the identification of the material was verified
Conclusions are correctly discussed, and related to the present findings

Additional comments

I appreciated the research, and surely can suggest to accept the paper in the present form

Reviewer 2 ·

Basic reporting

previous comment: - The authors cite two papers previously published by them:
Alves VM, Hernández MI, Lobo JM. 2018. Elytra absorb ultraviolet radiation but transmit NIR radiation in Neotropical Canthon Species (Coleoptera, Scarabaeinae).
Photochemistry and Photobiology 94:532-539.
Amore V, Hernández MIM, Carrascal LM, Lobo JM. 2017. Exoskeleton may influence the internal body temperatures of Neotropical dung beetles (Col. Scarabaeinae). PeerJ
5:e3349.
These two papers have similar hypotheses and methodology with the submitted manuscript, only other insect species were analysed. I would like the authors to discuss this prior work of theirs and to state how their current manuscript advances beyond or correlate with this prior work.

authors' respons: I am sorry but we disagree. In the first of these two mentioned papers the thermal ability of the beetle exoskeleton of some Geotrupidae species to regulate internal body temperature is examined subjecting dried specimens to several light sources. At this moment we still did not have access to a spectrometric device and the obtained results suggested that the variability in heating rates was mainly due to physical parameters as body volume. Subsequently, we continue the research in this field using a spectrophotometer on five Neotropical Canthon species (Scarabaeidae). The results provided by this study indicated that all the species showed a similar pattern in which the light from shorter wavelengths and higher frequencies is almost entirely absorbed by the elytra, while radiation from longer wavelengths and lower frequencies can mostly pass through the elytra. In this paper we suspect that one of the considered species (C. quinquemaculatus) might have a differential response linked to its ecological preferences. This possibility has motivated us to examine this question in other phylogenetic close and sympatric species with clear discrepant ecological preferences. The results provided in our study constitute a natural follow-up to examine the probable association between the ecological preferences of different species and the behaviour of the elytra under different radiations. So, it is obvious that the three papers follow the same research line but do not address the same issues, nor address them in the same way.

new comment: i suggest a shortened version of the comparison of the previous and this paper (a part of your answer) to be included in the manuscript.

Experimental design

no new comments

Validity of the findings

Deleted statement from the discussion (lines 404-406, track changes word version):

"However, Pantelić et al.,(2017) talk about the low capacity of melanin in the absorption of the IR spectrum in a dusk moth species, which could indicate that the structural component in which the pigment is embedded could influence its response"- nicely correlates with the rest of the discussion, and explains that the structure could have a high influence to reflection/absorption properties of the elytra, and I suggest keeping it in the manuscript disscussion.

Additional comments

The authors have adequately addressed my concerns in their revision. I have just two more suggestions, mentioned above. After that I support acceptance of this manuscript for publication.

Reviewer 3 ·

Basic reporting

No comment.

Experimental design

No comment.

Validity of the findings

No comment.

Additional comments

The paper has improved and now is clear and unambiguous. The authors present new and relevant results on the physical mechanisms responsible for the management of environmental radiation by the exoskeleton of these beetles. I have only one stylistic correction.

Line 428.- Author names and year in bold.

---

## Round 0.3 · accepted · Accept

Dear Drs. Cuesta and Lobo:

Thanks for re-submitting your revised manuscript to PeerJ, and for addressing the concerns raised by the reviewers. I now believe that your manuscript is suitable for publication. Congratulations! I look forward to seeing this work in print, and I anticipate it being an important resource for research communities studying the ecology and evolution of beetles.

Thanks again for choosing PeerJ to publish such important work.

-joe